# The Resemblance between Bacterial Gut Colonization in Pigs and Humans

**DOI:** 10.3390/microorganisms10091831

**Published:** 2022-09-14

**Authors:** Michiel Van de Vliet, Marie Joossens

**Affiliations:** Laboratory of Microbiology, Department of Biochemistry and Microbiology (WE10), Faculty of Sciences, Ghent University, 9000 Ghent, Belgium

**Keywords:** gut microbiota, intestinal colonization, human, pig, niche dependency

## Abstract

Thorough understanding of the initial colonization process of human intestines is important to optimize the prevention of microbiota-associated diseases, and also to further improve the current microbial therapies. In recent years, therefore, colonization of the human gut has gained renewed interest. However, due to a lack of standardization of life events that might influence this early colonization process in humans, many generally accepted insights are based on deduction and assumption. In our review, we compare knowledge on colonization in humans with research in piglets, because the intestinal tract of pigs is remarkably similar to that of humans and the early-life events are more standardized. We assess potential similarities and challenge some concepts that have been widely accepted in human microbiota research. Bacterial colonization of the human gut is characterized by successive waves in a progressive process, to a complex gut microbiota community. After re-analyzing available data from piglets, we found that the bacterial colonization process is very similar in terms of the wave sequence and functionality of each wave. Moreover, based on the piglet data, we found that, in addition to external factors such as suckling and nutrition, the bacterial community itself appears to have a major influence on the colonization success of additional bacteria in the intestine. Thus, the colonization process in piglets might rely, at least in part, on niche dependency, an ecological principle to be considered in the intestinal colonization process in humans.

## 1. Introduction

The human gut community is a complex, dynamic microbial network comprising of bacteria, yeast, and viruses. Due to its relevance to the host’s health and the multiple physiological functions it performs, it is sometimes referred to as the hidden organ of the body. The intestinal microbiota assists in digestion of insoluble fibers, thereby providing nutrients (short-chain fatty acids, SCFAs) for the colonic epithelial cells [1]. Besides its digestive function, it also protects the host from pathogens by occupying the available niches (space and nutrients) and plays a crucial role in the development of the host’s immune system [2,3]. The latter function is thought to be heavily impacted by the developing gut microbiota during the first years of life. The adult human colon is densely populated by bacteria with estimated numbers being as high as 10^14^ bacterial cells per gram in the luminal content [1]. The massive colonization of the intestines is assumed to start gradually from birth onwards until the dynamically more stable adult state is reached. Interestingly, disruptions in this colonization process have been shown to coincide with increased incidences of allergic diseases during childhood [4,5].

Regardless of the continuing research, our knowledge on intestinal colonization in humans is far from complete. Major knowledge gaps appear to result from the lack of datapoints per infant, which necessitates deduction-based conclusions and hampers in-depth insight in succession and colonization of specific bacteria in the intestinal microbial network. As dense sampling and high-quality collection of fecal samples of newborns is not straightforward for humans and key factors affecting the microbiota development (major dietary and environmental changes) are not standardized, we here summarize the colonization of the porcine gut, in comparison with what is known in infants. After all, anatomically and physiologically speaking, pigs closely resemble humans [6]. In addition, early development of the porcine gut microbiota community is more standardized, given the more consistent duration of lactation and co-habitation, for instance. For piglets, four subsequent growth phases are described, which are each well-defined, namely lactation, nursery, growing, and the final phase of weight increase named finishing. During the lactation stage, the piglet is co-housed with littermates and the mother sow. The piglets are then separated from the sow (weaning) and are co-housed with piglets of a similar age throughout the subsequent growth phases until they have reached their adult market weight. Due to these more standardized circumstances, projecting the knowledge of the porcine gut colonization on the human gut colonization might assist in a better understanding of the latter. In this review, we focus on bacterial gut colonization and assess to what extent the available data on the intestinal colonization of the porcine gut can help to fill the knowledge gaps in intestinal colonization of humans.

## 2. State-of-the-Art on Human Bacterial Gut Colonization

Intestinal bacterial colonization in humans has been established as a sequence of different bacterial waves that support and replace each other. As most data are available for vaginally delivered breastfed infants, we summarize the state of the art based on data from those children. The process of developing a complex microbial community has shown to display consistent, distinguishable colonization waves which succeed each other. In humans, subsequent waves of colonization occur over the course of the first two years of life, leading to a more stable and adult-like microbiota around the age of 2 years [3]. While it was formerly proposed that bacterial colonization already started in utero [7], this has been refuted by recent studies [8,9]. Therefore, it is now generally accepted that newborns encounter their first bacteria during birth. As such, from all bacteria that the newborn encounters at birth, the first stable colonizers (first colonization wave) in humans comprise mostly facultative anaerobes such as enterobacteria and bacilli such as *Streptococcus*, *Enterococcus*, and *Staphylococcus* [10,11,12,13]. These bacteria remove the oxygen from the gut and thereby set an important prerequisite for the next colonization waves, which comprise strict anaerobes [14,15,16].

Once anaerobic, the genus *Bifidobacterium* (second colonization wave) expands rapidly, becoming the dominant member of the gut microbiota of the newborns with an exclusive milk-based diet at that time. *Bifidobacterium* degrades mucin and human milk oligosaccharides (HMOs), a major constituent of breastmilk. The molecules resulting from degradation of HMOs can then be further metabolized by other bacteria that cannot or only partly metabolize HMOs, a mechanism called cross-feeding [17]. As such, *Bifidobacterium* in turn helps to create a favorable environment for colonizers of the third wave, which comprises of *Bacteroidota* and other strict anaerobic bacteria belonging to the *Clostridiales*. Additionally, several butyrate-producing bacteria such as *Anaerostipes*, *Roseburia*, and *Faecalibacterium* are described to become part of the infant gut’s bacterial community at that time [13,18,19]. The transition from second to third wave appears to co-occur with weaning (when the infant is no longer breastfed) [10]. Some publications [12,18] also describe a distinct fourth colonization wave which comprises the further increase in *Clostridiales*, more specifically, *Lachnospiraceae* and *Ruminococcaceae*. The latter has been related to a shift in diet enriched in plant-based products [10], which is in line with data on the impact of dietary interventions on adult microbiota composition [20].

Some factors, however, impact this colonization pattern. The delivery mode is the most important factor, the one with the most profound repercussion on the gut microbiome development [21]. During vaginal birth, a newborn encounters maternal faecal, vaginal, and skin bacteria. While not all these bacteria have the ability to establish themselves in the infant gut [10], this microbial exposure at the start is markedly different from birth via caesarean section (CS), which is performed in a sterile manner. In contrast to vaginally born infants, infants born via CS have a lower abundance of *Bifidobacterium* and an almost complete lack of *Bacteroidota*. Furthermore, CS infants often have a higher abundance of *Klebsiella oxytoca*, *Klebsiella pneumoniae*, *Enterobacter cloacae*, *Enterococcus faecium*, *Staphylococcus aureus*, and *Clostridium perfringens*, several of which are typically hospital-associated bacteria and potential pathogens [21,22,23,24]. CS-associated alterations in the gut microbiota composition are most pronounced up to 6 months after birth, but even a much more prolonged effect on the gut microbiota is suspected [10]. In a large adult population (*n* = 1106), compositional differences based on the delivery mode could, however, no longer be identified [25], but long-term health effects are still under investigation. Additionally, formula feeding, resulting in a decrease in *Bifidobacterium*, and antibiotic use, resulting in an increase in facultative anaerobes such as *Escherichia* and *Klebsiella*, largely influence the developing gut microbiota compared to the colonization process described for breastfed, vaginally delivered babies [21]. 

Cooperation and dependence between different waves have been comprehensively described, yet this knowledge remains insufficient as it is deduced from interval sampling. The timing and length of the different colonization waves differ between infants, even when major colonization routes are similar (delivery mode, breastfeeding, etc.) [13]. With only limited timepoints, it is very difficult to reconstruct and understand the biological grounds for these differences, as a much more detailed view on the gut microbiota transitions would be required. However, since pigs have a more standardized course of their lives during the first weeks (birth, suckling, co-habitation), they might prove an elegant solution to learn how external factors influence the development of the gut microbiota, and more specifically, which events are inextricably linked to changes in the gut microbiota composition.

## 3. Intestinal Colonization in Porcine: Re-Analyses of Existing Data

We here assessed to what extent gut colonization in pigs can help to fill the gaps in humans. Over the years, many studies have been conducted on the gut colonization in pigs, yet the majority of these studies had a different focus for their research. On the one hand, there are studies that compared the bacterial community in response to an antibiotic treatment (case vs. control studies), while on the other hand, there are studies focusing on the influence of weaning on the gut microbiota and the effect of different types of feed on the post-weaning community. For both cases, although samples were collected over time and the bacterial community was described for all timepoints, the focus of the research was directed towards the influence of the confounding factor on the gut microbiota rather than the evolution of the gut microbiota over time. The focus in this review lies, however, on the developing gut microbiota; therefore, to allow a proper comparison between human and porcine bacterial gut colonization patterns, existing data on piglets are re-analysed with the purpose to build, for the first time, a colonization model following the example of the already existing human colonization models.

### 3.1. Study Selection

Using the software program Rayyan [26], studies on the intestinal colonization of piglets were filtered after a wide search in PubMed using following search terms: porcine gut colonization; swine gut colonization; pigs gut maturation; swine gut maturation; piglets gut colonization; porcine intestinal colonization; bacterial gut colonization in pigs; porcine gut microbiota. Studies were included if they had multiple sampling-points over a defined time period and if no perturbations (antibiotics, FMT,…) were reported that could alter the natural intestinal colonization process.

To facilitate the re-analysis and to reduce technical noise, papers using a similar sequencing strategy were selected, more specifically studies targeting the V4 region of the 16S gene, using paired-end reads and next-generation sequencing technologies. As such, three studies for which raw reads and accompanying metadata could be extracted from the European Nucleotide Archive (ENA) were used for data re-analysis as described further, encompassing a total of 52 piglets and 31 timepoints (Accession numbers of the included studies: PRJNA613280 and PRJNA531671) [27,28].

Upon retrieval of the raw sequencing data, DADA2 pipeline was used for quality filtering, pairing of the reads and chimera removal in R (version 4.2.1) [29]. Reads were truncated at 240 and 160 base-pairs for forward and reverse reads, respectively. Taxonomic classification was performed using the SILVA rRNA database version 138.1 [30]. Graphs were made using the R-packages phyloseq and ggplot2 and via GraphPad Prism 9.

### 3.2. Results

The collected data on intestinal colonization of piglets covered thirty-one timepoints that were sampled on the day of birth and during four subsequent growth phases (lactation–nursery–growing–finishing). In line with what is known for humans, a divergent pattern is observed regarding alpha- and beta-diversity at birth. While a high alpha-diversity is observed for day 0, this dramatically drops at day 1 to reach the lowest observed alpha-diversity over time (Figure 1A). From then on, the alpha-diversity increases rapidly during the lactation phase and eventually keeps gradually increasing during the nursery phase and later stages. 

When assessing the beta-diversity of the samples, using Bray–Curtis diversity plots (Figure 1B,C), a clear sequence of the samples is seen, in line with their age. However, again, the day of birth is an outlier. Figure 1B,C show that the day of birth clusters closer to the later stages (growing–finishing) than to day 1 and the lactation stage. Apart from the day of birth, the data show consistent clustering per stage (Figure 1B) and consistent patterns over the subsequent days (Figure 1C), suggesting that, similar to what has been described for human infants, there is a clear pattern over time, indicating a gradually developing bacterial community.

As said, day 0 is atypical compared to the other days at the beginning of the colonization process because it has a high diversity and is more similar to the gut microbiota composition of older ages (growing and finishing phase) than younger ages (lactation phase). When looking at the taxonomic distribution, some species present at day 0 (*Ruminococcus*, *Turicibacter*, and *Terrisporobacter*) only re-appear in the top 20 abundant genera in the growing or finishing stage (Figure 2). These results can be explained by the fact that at birth, the piglet is exposed to the adult microbial communities (vaginal, gut, skin) of the sow, explaining the high alpha-diversity at birth. Since finishing pigs start to resemble an adult-like gut microbiota, this explains the proximity of day 0 community to the late phases on Bray–Curtis distance plots. As both the diversity as well as the community drastically change compared to the subsequent day (day 1), most of the bacteria the piglet encounters are transient by-passers, unable to stably colonize from their first passage in the gut [31].

The first days after birth, the intestine is dominated by facultative anaerobes, in particular *Escherichia/Shigella* (Figure 2). Within the first week (days 1–7), these facultative anaerobes are replaced with *Bacteroides*, which now becomes the dominant genus in the gut microbiota community. During the lactation phase, in addition to *Bacteroides*, the gut is colonized by *Fusobacterium*, which reaches its highest abundance at day 7 and then again decreases in abundance; and *Lachnoclostridium*, which remains relatively stable until weaning. Weaning marks the end of the lactation phase and the start of the nursery phase, which is promptly reflected in the gut microbiota composition. Although *Bacteroides* still constitute a large part of the gut microbiota the first two days after weaning (day 22–23), their proportion decreases sharply the third day after weaning (day 24). Simultaneously, the abundances of *Prevotella*, *Megasphaera*, and *Lactobacillus* increase during these first days after weaning, becoming the dominant members of the gut microbiota during the nursery phase (Figure 2). Additionally, SCFA producers such as *Faecalibacterium* and *Subdoligranulum* are now appearing among the most abundant members of the gut microbiota. Remarkably, while during lactation, along with *Lactobacillus*, *Ligilactobacillus* and *Limosilactobacillus* are highly abundant, the latter two do not appear anymore in the top 20 abundant genera after lactation. During the growing phase, a further increase in obligate anaerobes and *Streptococcus* is observed. *Clostridium sensu stricto* re-appears during this phase after its disappearance from the top 20 most abundant genera in the nursery phase. Even more, *Clostridium* abundances increase rapidly during the finishing phase, with coincides with a decrease in the abundance of *Prevotella*.

As such, we identified four different colonization waves, leading the porcine gut microbiota up to its adult state. These four colonization waves only partly overlap with the growth phases and thus weight increase in the piglets. Both initial waves take place during the lactation phase. The first colonizers are facultative anaerobes (first wave), mostly enterobacteria followed by a *Bacteroides* wave (second wave) that coincides with suckling. After weaning, when solid food is introduced, this wave is succeeded by a *Prevotella* wave (third wave) that is eventually succeeded by a *Clostridium* wave (fourth wave). With every wave, the gut microbiota becomes more complex until an adult state, characterised by a high diversity and functional redundancy, is reached. The occurrence of these waves, although not formally named like this, is corroborated by published data, where an initial facultative anaerobic dominance is also described, followed by a pre-weaning microbiota dominated by *Bacteroides* and to lesser extent *Enterobacteriaceae*, *Fusobacterium*, and *Enterococcus*, and post-weaning microbiota dominated by *Prevotella* and in lesser extent *Megasphaera*, *Lactobacillus*, and *Faecalibacterium* [32,33,34,35,36]. However, *Ruminococcus* and *Roseburia* have also been indicated as dominant gut microbiota members post-weaning, which was not observed in the re-analysed data here. The higher abundance of *Clostridium sensu stricto* at the beginning of colonization was also recorded earlier [33,35], as well as its high abundance in finishing pigs [37,38]. 

Furthermore, our observation that the increase in *Clostridium* at older age was accompanied with a decrease in the dominant *Prevotella* genus is in line with the paper from Han and colleagues [38], who even showed a negative correlation between these two genera at older age. Interestingly, Liu and colleagues [39] identified two co-excluding enterotypes in pigs, and named them after their main driver, namely *Prevotella* and *Treponema* enterotype. In our re-analyses of existing data, this competition appears to be reflected likewise, since *Treponema*, which is present starting from weaning, reaches its highest abundance when *Prevotella* is at its lowest, right after weaning and during the fourth colonization wave. Interestingly, *Bacteroidales* dominate the intestinal gut microbiota of pigs during the largest part of their development (first *Bacteroides* during lactation followed by *Prevotella* post-weaning).

Most studies on the initial colonisation of the piglet intestines have been targeted to a specific research question. Nevertheless, when going through the published data, it became apparent that in piglets, different bacterial taxa were also alternating dominant at specific timepoints in the maturation of the complex intestinal microbiota. When re-analysing the data from a selection of papers using a similar sequencing strategy, we revealed a wave-like pattern of succeeding bacterial taxa, similar to what has been described for humans.

## 4. Comparison between the Human and Porcine Bacterial Gut Colonization

The current literature on the initial colonisation of the piglet intestines, substantiated by our data re-analyses, supports the existence of subsequent waves, similar to humans, leading up to a complex intestinal microbiota. In line with what has been described in humans, a first wave of facultative anaerobic bacteria colonizes the intestines of new-born piglets (Figure 2). Indeed, the same wave has been observed in humans, where it is believed to create an anaerobic niche allowing the next colonisation waves to happen [14,15,16]. However, the subsequent wave observed in humans differs from the one observed in pigs. While in humans the second wave is dominated by *Bifidobacterium*, coinciding with exclusive (breast)milk feeding [10], in pigs, this subsequent wave is dominated by *Bacteroides*. Despite this taxonomic difference in the wave composition between humans and pigs, the functionality of this second wave of intestinal colonisers appears to be alike. After all, similar to *Bifidobacterium* in humans, *Bacteroides* serves to break down the complex milk oligosaccharides in pigs [40]. In addition, *Bacteroides* has likewise been identified as a primary degrader, capable of cross-feeding with other SCFA-producing species such as *Faecalibacterium prausnitzii* [41,42], *Anaerostipes caccae* [43], *Eubacterium* [44], and *Subdoligranulum* [45] and within the *Bacteroides* genus [46]. The latter was also illustrated by Rakoff-Nahoum [47] and colleagues, who demonstrated for *Bacteroides*
*ovatus* specifically that the cross-feeding capacity increased this species’ fitness thanks to reciprocal benefits from gut bacteria, among which is *Bacteroides vulgatus*, feeding on the inulin breakdown products. 

Weaning, and the concurrent change in diet, appears to initiate the next colonisation wave both in humans and pigs. As milk glycans are replaced with plant-based glycans, bacteria that can degrade the latter, such as *Prevotella*, have a competitional advantage, which is reflected in their increased abundance [33,48]. Despite the fact that pigs are omnivores, their feed after weaning is exclusively plant-based. This might explain why, in contrast to humans, the third colonisation wave in piglets is clearly dominated by *Prevotella* (Figure 2). Nevertheless, in humans, an increase in bacteria competent to degrade complex plant-based polysaccharides is likewise observed in the third colonization wave. In addition, this wave is enriched by different taxa that have butyrate-producing potential such as *Faecalibacterium* and *Roseburia*. Based on the current literature, this wave appears to be functionally in agreement with but is taxonomically less-well pronounced as compared to piglets, which is reflected by a combination of *Clostridia* and *Bacteroides* defining the third wave in the model for humans (Figure 3). An increasing complexity of the developing intestinal microbial community is further evident for both humans and pigs. Therefore, the fourth wave is also remarkably similar in both with a further increase in species belonging to the *Clostridiales*.

Two important things stand out in these last waves. First, while in humans the transition from the third to fourth wave is thought to result from a change in diet, this can be called into question based on this data in pigs. After all, in pigs, weaning and the introduction of solid food happen at the same moment, while in humans these are two separate key events, often with considerable time in between. Second, in humans, *Clostridiales* are dominant from weaning onwards, suggesting a potential causal link between both. However, in pigs, the *Clostridiales* only become prominent from the fourth wave onwards. As such, this might imply a link to microbial community maturation rather than to the event of weaning. In other words, the stable integration of *Clostridiales* in the intestinal microbiota potentially requires a more complex network and the insurance of availability of specific functions through functional redundancy. 

## 5. How Unique Is This Bacterial Colonization Process?

Despite some differences in taxonomic composition between the human and porcine colonization waves, it appears that the functionalities of the waves are well preserved. Earlier work [49] focusing on the porcine gut microbiota network showed that the ecological community interactions and functionality of the developing gut microbiota between humans and pigs are preserved, confirming our observations. This remarkable trait raises the question of whether the bacterial colonization process, as described in this review, might be preserved among the *Mammalia*. To answer this question, this required additional data about the gut microbiota colonization of another mammalian species, belonging to a different order than humans (*Primates*) and pigs (*Artiodactyla*) do. To the best of our knowledge, there are no additional data on the early gut colonization in classical experimental animals such as mice, rats, and rabbits. However, some data are available for another mammal, namely dogs (belonging to the order *Carnivora*), which we here summarize shortly. Although a limited amount of data is available regarding gut colonization in dogs, it does seem to follow the same pattern as humans and pigs. Shortly after birth, *Enterobacteriales* and *Clostridiales* are abundant, but the pre-weaning gut microbiota evolves to a community consisting, among others, of *Bacteriodales*, *Lactobacillus*, *Bifidobacterium*, and *Fusobacterium*. With aging, the community becomes richer in members belonging to *Clostridiaceae*, *Ruminococcaceae, Lachnospiraceae*, and *Fusobacterium* while pre-weaning genera such as *Bifidobacterium* and *Bacteriodales* typically decrease [50,51,52,53].

In addition to the data found in dogs, one study assessed the bacterial intestinal colonization in cats [54], another member of the order *Carnivora*. Here, the microbial community was monitored after weaning, which typically occurs at around 12 weeks of age in kittens. Three sampling times were assessed in 30 kittens with the first at a mean age of 16 weeks, the second at a mean age of 28 weeks, and the third a mean age of 40 weeks. Here as well, the gut microbiota community was shown to evolve from an early stage dominated by *Bifidobacterium* and *Lactobacillus* (mean age of 16 weeks) to a gut microbiota dominated by *Bacteroides, Prevotella*, and *Megasphaera* (mean age of 40 weeks). Furthermore, an increasing alpha-diversity (Shannon index) was observed over time, while the beta-diversity displayed a time-dependent pattern, similar to our own data [54]. Taken together, this again points towards the gradual development of the gut microbiota. Remarkably, while cats normally wean around 12 weeks, and thus, the first sampling happened several weeks after weaning, the gut microbiota was still dominated by typical suckling-associated bacteria. This might again suggest that the different colonization waves are not entirely determined by external factors. An important remark here is that there are no earlier data available in this study, which makes it impossible to know what the gut microbiota composition during suckling was. This implies that although *Bifidobacterium* and *Lactobacillus* were still dominant after 18 weeks, they potentially reached much higher abundances during suckling and were then decreasing over time. In that case, the first sampling point (16 weeks) would fall during the transition of two different colonization waves.

The development of the canine and feline gut microbiota seems to confirm the preservation of some general principles applying to early bacterial intestinal colonization among the *Mammalia*. However, natural intestinal colonization in other mammals should be studied in order to clearly confirm evolutionary preservation. In addition, it would also be interesting to compare mammalian gut microbiota colonization with non-mammalian gut microbiota colonization, for example, in chickens, to further confirm the (causal) link between external events and changes in the gut microbiota.

## 6. Colonization Relies on a Supportive Environment

The famous quote of Prof. Baas Becking, “Everything is everywhere, but the environment selects” [55], dating from 1934, is surprisingly relevant today and still applies to many environmental, complex bacterial communities. In essence, this simply means that the existing environmental state (nutrients, cross-feeding, microbial competition, etc.) defines the subsequent environmental state. Although this principle is very well-known in environmental microbiology, it is scarcely mentioned when describing complex microbial communities in biomedical microbiology. However, with the current data at hand, the principle seems to also apply on the intestinal colonization process.

Like humans, pigs are exposed at birth to bacteria from different sources such as the sow’s vaginal, skin, and faecal microbiota and bacteria from environmental sources such as the (slatted) floor [56,57]. As massive microbial seeding during natural birth does not immediately result in a stable and diverse microbial community, it is expected that the colonization of the human (and pig) gut happens in line with ecological principles, as have been described for other complex microbial habitats [58]. During lactation, the pigs are also exposed to the milk microbiome of the sow, although it appears that this source has a lesser influence on the pig’s gut microbiota compared with humans [32]. Strikingly, when evaluating the influence of different bacterial sources on the gut microbiota, it appears that shortly after birth the sow’s vaginal and environmental bacteria are the most important sources. It is only during lactation that the influence of the sow’s faecal microbiota starts to rise, which is suspected to result from co-occurrence with former settlers [56]. Similar results were observed in humans, where vertically transmitted species or milk-associated bacteria were only able to stably colonize long after the first encounter with the infant’s gut [12,59]. For example, some obligate anaerobic species that colonize only in the third or fourth wave (*Clostridium*, *Coprococcus*) are already present from the beginning in breastmilk [60], but apparently fail to colonize stably. The strict anaerobic strains for which vertical transmission has been shown (e.g., *Ruminococcus bromii*) [61] are also only detectable in the infant’s microbiota after several months. This points towards the need for niche preparation, in which bacteria have to be supported by the environment in order to colonize successfully. Interestingly, re-colonisation of the gut using faecal microbial transfer (FMT) in adults has been shown to depend on niche availability [62,63]. Here as well, stable implantation of donor species was shown not to be random but to depend on niche availability to allow long-term colonization.

These findings point towards niche dependency, i.e., the requirement for some bacteria or nutrients to be present before certain bacteria are able to colonize the gut. Niche dependency can provide an explanation for the drastic shift in gut microbiota community from day 0 to day 1, which shows a dramatic decrease in diversity from day 0 to day 1. Most of the bacteria encountered at day 0, previously called transient by-passers, do not find the required conditions for stable colonization and thus disappear. Since pigs are coprophagic, which means they consume their own, littermates’, and/or sow’s faeces, one would expect to retrieve more adult-associated bacteria. The absence of this observation further supports the hypothesis of niche dependency.

While niche dependency has always been thought of as nutrient availability, and thus directly related to the dietary regiments, this is starting to change. In 1983, Freter proposed the nutrient niche theory, in which the capability of a bacteria to efficiently use at least one limiting nutrient defined its ability to colonize the gastrointestinal tract [64]. However, this theory proved to be insufficient to describe a complex system such as the gut ecosystem. If only a single limiting nutrient dictates the gut population, a much more unstable gut microbiota would be expected due to daily-to-daily variations in diet [65]. Additionally, cross-feeding abilities by certain bacteria, such as the mucin-degrader *Akkermansia muciniphila*, contradict this theory [66]. Therefore, acquisition of a complex adult gut microbial community not only depends on the nutrient availability but also on the previously existing state of the gut microbiota [67,68]. Meng and colleagues [69] noticed that the intestinal epithelium of a populated gut contains a high proportion of fucosylated glycans (used as an attachment place by some bacteria) on its surface while an unpopulated gut does not. They showed that this fucosylation was a result of the colonization-stimulated ERK and JNK signaling pathways in epithelial cells, and that blocking of it could prevent further bacterial colonization. As such, pioneer bacteria create a supportive environment for the next generations of bacteria.

To further back this up, in-depth analyses of the functionality of the subsequent bacterial waves would be required. By unraveling the genetic and metabolic potential per wave, cross-feeding opportunities might be anticipated, helping to predict which novel niches are becoming available. 

## 7. What Can We Learn from Pigs about the Human Gut Colonization?

In an attempt to bridge some important knowledge gaps in the field of human intestinal colonization with bacteria, we re-analyzed existing data on piglets. Combining different sample sets, we were able to model the early colonization in pigs, using data from fifty-two individual animals during the first six months of their life. We demonstrated that, as in humans, distinct colonisation waves can be distinguished in piglets. Moreover, for each of the subsequent waves, the functionality appeared to be highly similar to the ones hypothesized in humans, and for both humans and piglets, this sequence of colonisation waves led to an increased complexity and adult-like composition of the intestinal microbiota. As such, the proposed wave model for intestinal colonisation in humans [10] was shown here to also apply for piglets (Figure 3). Moreover, the general principles behind this colonization model seem at first glance to have been preserved in mammals.

While several potential key events in complex microbial community assembly cannot easily be standardized in humans (duration of breastfeeding, interaction with siblings and adults, etc.), the farming practices for piglets allowed to analyze more standardized data from piglets. Our approach revealed some potential current misconceptions in human intestinal colonization. The transition from the third to the fourth colonization wave in humans been thought to be driven by a change in diet, while we see a similar transition in piglets for whom, at that time, the feed does not change. Likewise, the observed increase in *Clostridiales* in human infants has been suggested to result from weaning. In piglets, however, a larger timespan between weaning and a rise in *Clostridiales* was seen, indicating that the observed increase in *Clostridiales* in humans might also depend on more factors than weaning alone. These observations highlight the potential of animal data to refine human intestinal colonization models. After all, on top of the lack of standardisation, dense sampling in the early colonization phases in newborns is not obvious, as illustrated in current literature. 

To conclude, the assembly of a complex adult intestinal community is not random, and a striking overlap has been described between humans and pigs. Although these results derived from pigs can bridge some knowledge gaps in humans, we still have to keep in mind that they remain, despite many resemblances, different species. Therefore, the next step would be to perform in-depth analyses on the human gut microbiota derived from human samples. This, however, requires very dense, sequential sample collections, and until this requirement is met, studies on farm animals such as pigs are the next best thing to learn about human bacterial intestinal colonization.

## Figures and Tables

**Figure 1 microorganisms-10-01831-f001:**
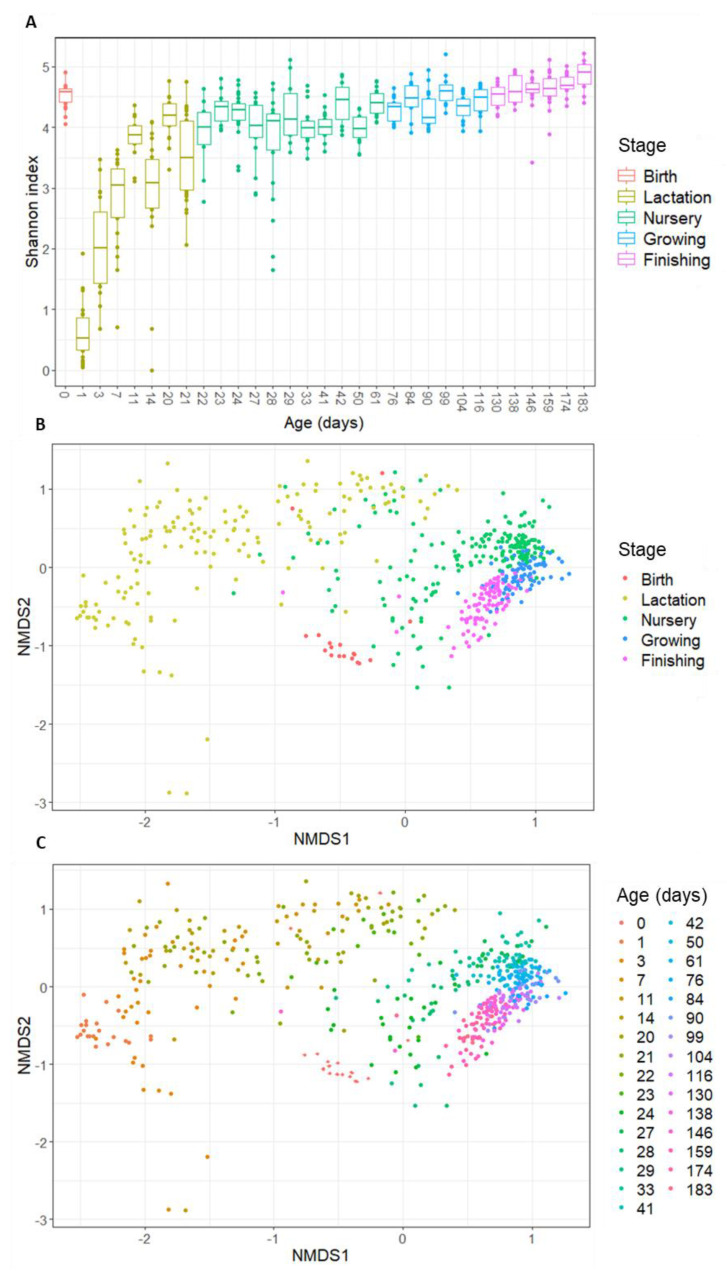
(**A**) Alpha-diversity (Shannon index) of the porcine gut microbiota in function of age, coloured per growth stage; (**B**) beta-diversity (Bray–Curtis distances) of the porcine gut microbiota in function of growth stage; and (**C**) beta-diversity (Bray–Curtis distances) of the porcine gut microbiota in function of age (in days). Samples from day 0 are depicted with a diamond shape while the other ages are depicted using dots in graph (**C**).

**Figure 2 microorganisms-10-01831-f002:**
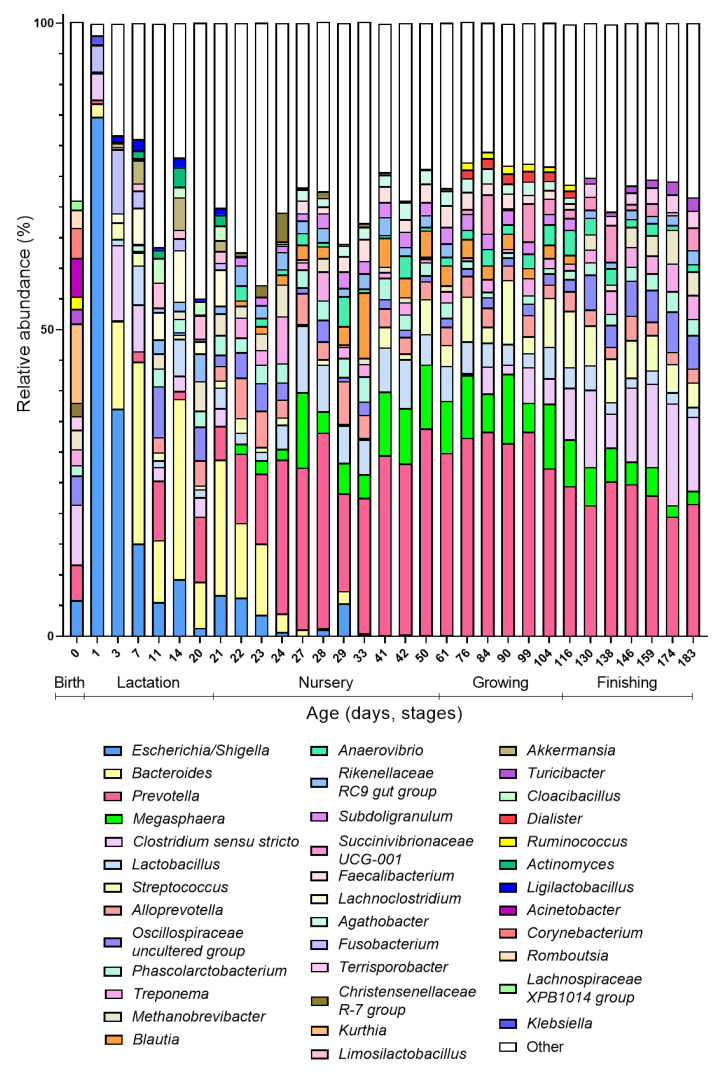
Taxonomic composition of the gut microbiota over time. Mean relative abundance (%) at genus level is plotted over time (day 0–day 183). Only the top 20 most abundant genera per growth stage with minimal 1% abundance were included in the graph. The different phases are indicated below the days on the x-axis (Birth: day 0, Lactation: days 1–21, Nursery: days 22–61, Growing: days 76–116 and Finishing: days 130–183).

**Figure 3 microorganisms-10-01831-f003:**
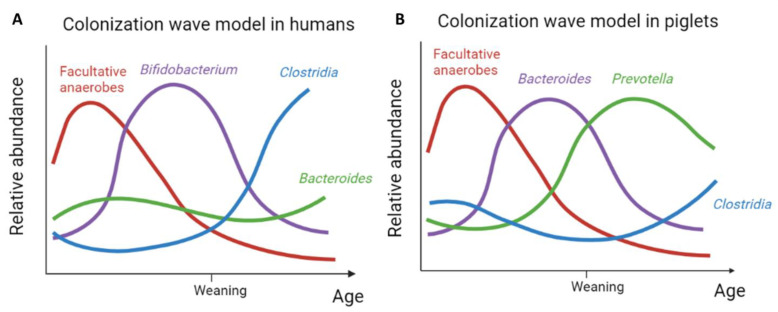
Model depicting the different bacterial colonization waves (not on scale) in humans (**A**) and pigs (**B**) over time. Our model in piglets was based on the colonization wave model in humans as proposed by Korpela and colleagues. Panel A: adapted with permission from Ref. [10] 2021, Korpela et. al.

## Data Availability

Data accession numbers for the studies used for the re-analysis can be found in the text, under Section 3.1.

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
