# Peer review of "The Resemblance between Bacterial Gut Colonization in Pigs and Humans"

_microorganisms, 2022, doi:10.3390/microorganisms10091831_

Round 1

Reviewer 1 Report

This is a sound and well-argued manuscript, a useful read for researchers in the field. The obervation of wave-like similarities is relevant. The authors may want to add a paragraph whether or not the waves mean something in terms of gene functions, like enzymes appearing/disappearing. This may be carried out by the same sw as was used for the species-wise descriptions. Even if the evaluation is not possible now, a paragraph on the views of the authors would be interesting for the readers. 

Author Response

Reviewer 1:

This is a sound and well-argued manuscript, a useful read for researchers in the field. The obervation of wave-like similarities is relevant. The authors may want to add a paragraph whether or not the waves mean something in terms of gene functions, like enzymes appearing/disappearing. This may be carried out by the same sw as was used for the species-wise descriptions. Even if the evaluation is not possible now, a paragraph on the views of the authors would be interesting for the readers. 

Thank you for your appreciation of our work. Throughout the review, we have tried to link the bacterial abundances with functionality and metabolic potential. However, we feel that solid data is currently missing to state anything more substantial on that topic. Upon the Reviewer’s request, we have added this to the manuscript (lines 596-599):

“To further back this up, in-depth analyses of the functionality of the subsequent bacterial waves would be required. By unraveling the genetic and metabolic potential per wave, cross-feeding opportunities might be anticipated, helping to predict which novel niches are becoming available.”

Reviewer 2 Report

The present manuscript focus on the resemblance between bacterial gut colonization in pigs and humans.  The subject frame of the work is well constructed. So, in this respect and this article should be contributed to present research. I recommended this work for publication after the following minor revisions.

1.      There are several typographical mistakes as well in whole manuscript. Therefore, the author’s thoroughly careful check the language and typo mistake to minimize the error.

2.      The abstract should be beginning with a sentence about the background of concept and the aims as well as novelty of study should be mentions. What exactly is the novelty of this study? The abstract is poorly written and should be improved. Abbreviations must be avoided in abstract. Parenthesis should be avoided in abstract - this is poor writing. Please improve.

3.      All figures are of poor technical quality and not suitable for publication, especially in a high reputed journal. Font size and kind is too small and must be unified in all figures. Small writings are unreadable. All figures must be self-explanatory. Axis titles are poorly presented or absent. Units are missing. Are the data presented in figures significantly different? At least error bars should be shown.

Author Response

 Reviewer 2:

The present manuscript focus on the resemblance between bacterial gut colonization in pigs and humans.  The subject frame of the work is well constructed. So, in this respect and this article should be contributed to present research. I recommended this work for publication after the following minor revisions.

  1. There are several typographical mistakes as well in whole manuscript. Therefore, the author’s thoroughly careful check the language and typo mistake to minimize the error.

In line with the Reviewer’s remark, we have corrected language mistakes and typos throughout the manuscript. They are highlighted in the revision. The mistakes on lines 222, 230, 241, 253, 290, 353, 356, 357 and 388 have been corrected among others.

  1. The abstract should be beginning with a sentence about the background of concept and the aims as well as novelty of study should be mentions. What exactly is the novelty of this study? The abstract is poorly written and should be improved. Abbreviations must be avoided in abstract. Parenthesis should be avoided in abstract - this is poor writing. Please improve.

We have re-written the abstract and added the following introductory part to clearly state our aims for this review:

‘Thorough understanding of the initial bacterial colonization process of human intestines is important to optimize prevention of microbiota-associated diseases, and also to further improve now-a-days microbial therapies. In this review, we therefore summarize the most recent insights in the field based on groundbreaking studies. Alongside, we re-analyze data from the bacterial colonization process of piglet intestines to assess potential similarities and to challenge some colonization concepts that became generally accepted in human microbiota research.’

Additional changes are again highlighted in the revised abstract.

  1. All figures are of poor technical quality and not suitable for publication, especially in a high reputed journal. Font size and kind is too small and must be unified in all figures. Small writings are unreadable.

Thank you for pointing out the differences in font sizes for the Figures in our manuscript. This has been corrected now. Also, the technical quality of the Figures has been further improved upon the Reviewer’s request.

All figures must be self-explanatory. Axis titles are poorly presented or absent. Units are missing.

Based on the Reviewer’s remarks about the lack of clarity, lack of axis titles and units, we understood that the model in Figure 3 was not well-explained.  We had made this Figure based on an earlier representation in humans as explained in the legend. However, we agree that it was not clear as such. We therefore added a caption to make explicit that these two graphs are representing models for the consecutive prominent groups of bacteria in the gut during colonization. Since the abundances of the waves are not scaled, as also reported in the legend, no units are displayed for this graph either. After all, the relative heights of the waves are secondary to depict the wave-pattern.

Are the data presented in figures significantly different? At least error bars should be shown.

The aim of the current review was not so much to reproduce the original findings on significant differences between stages, as this has been reported in the initial papers. We aimed here at looking for similarities with what had been described in humans with respect to the colonization patterns.

We hence provided the error bars only in the figure where we believe it was most relevant, Figure 1A. For Figure 3 specifically, error bars cannot be added as it represents a non-scaled model.